# Use of an algorithm based on routine blood laboratory tests to exclude COVID-19 in a screening-setting of healthcare workers

Math P. G. Leers[1]ʘ*, Ruben Deneer[2,3]ʘ, Guy J. M. Mostard[4], Remy L. M. Mostard[5], Arjen-Kars Boer[2], Volkher Scharnhorst[2], Frans Stals[6], Henne A. Kleinveld[1], Dirk W. van Dam[6]

1 Department of Clinical Chemistry & Hematology, Zuyderland Medical Center, Heerlen-Sittard, The Netherlands, 2 Department Of Clinical Chemistry & Hematology, Catharina Hospital, Eindhoven, The Netherlands, 3 TU/E (Technical University), Eindhoven, The Netherlands, 4 Department of Internal Medicine, Zuyderland Medical Center, Heerlen-Sittard, The Netherlands, 5 Department of Pulmonology, Zuyderland Medical Center, Heerlen-Sittard, The Netherlands, 6 Department of Medical Microbiology & Infectionprevention, Zuyderland Medical Center, Heerlen-Sittard, The Netherlands

ʘ These authors contributed equally to this work.
* mat.leers@zuyderland.nl

**Data Availability Statement:** Due to legal restrictions on sharing the de-identified data, individual level data cannot be shared publicly.

## Abstract

### Background

COVID-19 is an ongoing pandemic leading to exhaustion of the hospital care system. Our health care system has to deal with a high level of sick leave of health care workers (HCWs) with COVID-19 related complaints, in whom an infection with SARS-CoV-2 has to be ruled out before they can return back to work. The aim of the present study is to investigate if the recently described CoLab-algorithm can be used to exclude COVID-19 in a screening setting of HCWs.

### Methods

In the period from January 2021 till March 2021, HCWs with COVID-19-related complaints were prospectively collected and included in this study. Next to the routinely performed SARS-CoV-2 RT-PCR, using a set of naso- and oropharyngeal swab samples, two blood tubes (one EDTA- and one heparin-tube) were drawn for analysing the 10 laboratory parameters required for running the CoLab-algorithm.

### Results

In total, 726 HCWs with a complete CoLab-laboratory panel were included in this study. In this group, 684 HCWs were tested SARS-CoV-2 RT-PCR negative and 42 cases RT-PCR positive. ROC curve analysis showed an area under the curve (AUC) of 0.853 (95% CI: 0.801–0.904). At a safe cut-off value for excluding COVID-19 of -6.525, the sensitivity was 100% with a specificity of 34% (95% CI: 21 to 49%). No SARS-CoV-2 RT-PCR cases were missed with this cut-off and COVID-19 could be safely ruled out in more than one third of HCWs.

However, these data will be made available to individuals who want access for scientific and/or academic research purposes and are willing to commit to handling the data in a manner which is consistent with confidentiality requirements. The criteria for access would broadly incorporate requests from individuals with credible academic/ research credentials. Requests for access to the data should be directed to the Ethics Committee of Zuyderland Medical Centre at METC@zuyderland. nl.

**Funding:** The author(s) received no specific funding for this work.

**Competing interests:** The authors have declared that no competing interests exist.

## Conclusion

The CoLab-score is an easy and reliable algorithm that can be used for screening HCWs with COVID-19 related complaints. A major advantage of this approach is that the results of the score are available within 1 hour after collecting the samples. This results in a faster return to labour process of a large part of the COVID-19 negative HCWs (34%), next to a reduction in RT-PCR tests (reagents and labour costs) that can be saved.

## Introduction

Severe acute respiratory syndrome coronavirus 2 (SARS-CoV-2) disease 2019 (COVID-19)) is an ongoing pandemic with at present over 150 million of cases and over three million deaths worldwide [1]. The initial clinical symptoms for COVID-19 are nonspecific and similar to other seasonal viral diseases, which encompass fever, dyspnoea, dry cough and fatigue. Many countries, including the Netherlands, are struggling to control COVID-19 outbreaks, especially in the detection of silent infections in the pre- or asymptomatic patient that can contribute to transmission [2]. Empirical studies have indicated that individuals may be highly infectious during the presymptomatic phase [3].

Healthcare workers (HCWs) potentially experience greater risks for emerging infectious diseases [4,5] due to occupational exposure to sick patients and virus-contaminated surfaces [6]. Contagious HCWs may infect patients, co-workers and family members. However, the withdrawal of ill HCWs from duty can threaten essential healthcare staffing during an epidemic [7]. Therefore, infection prevention and quick, accurate diagnosis of potential COVID-19 in HCWs are crucial to maintain hospital operations [8].

Consequently, understanding the prevalence of, and factors associated with SARS-CoV-2 infection among frontline HCWs who care for COVID-19 patients are important to protect both HCWs and their patients. Next to this, modelling analyses show that rapid case identification of infected persons is critical to interrupt transmission, especially for infectious cases without clinical symptoms [2]. At the moment, the health care system has to deal with a high level of absenteeism of HCWs with COVID-19 related complaints, and in whom an infection with SARS-CoV-2 has to be ruled out before they can return back to work.

Reverse Transcription polymerase chain reaction (RT-PCR) based methodologies are the gold standard in confirming that the individual presenting with COVID-19 has active viral shedding of SARS-CoV-2 [9]. However, there are some important limitations to RT-PCR. First, current techniques take up to 6–8 hours in order to obtain first results. Next to this, laboratories often cannot handle the overload of tests. A third important limitation is that RT-PCR on a nasopharyngeal swab, may be false negative in the initial phase of the disease, in spite of the presence of typical symptoms [10–12]. In addition, the standard test used has an 80% accuracy (compared to chest CT scan results) [12], which may depend on the specific level of viral shedding by any individual at the time of sample test. Fourth, the RT-PCR technique carries a certain cost, which could mean a considerable financial burden [13].

Very recently, an algorithm was developed by Boer et al, leading to the so-called 'CoLab'-score [14]. The score is calculated using 10 numeric values of routine-laboratory parameters next to the age of the patient. The linear predictor of the CoLab-score is continuous, therefore a cut-off can be chosen such that a high sensitivity and high negative predictive value can be achieved. This algorithm was developed and validated to exclude COVID-19 in patients presenting at the Emergency Department (ED).

The aim of the present study was to investigate if the CoLab-score could be used to exclude within one hour COVID-19 in a screening setting of healthcare workers, who requested a SARS-CoV-2 RT-PCR test because of COVID-19 related complaints, or because they were in close proximity to a SARS-CoV-2 infected person.

## Materials & methods

### Study design and selection of healthcare workers (HCWs)

We conducted a prospective screening study to assess the comparability between naso-/oro-pharyngeal swabs and the CoLab-score (based on routine blood tests). Healthcare workers were included during the period from January 2021 till March 2021 either:

- because of COVID-19 related complaints or

- because they were in close proximity to a person with COVID-19.

HCWs were required to have complete data on clinical chemistry and hematologic parameters, needed to calculate the CoLab-score. From the validation study [14] it is known that there are some external factors influencing the predictive value of the score. For this reason, the following HCWs were excluded from this study:
HCWs with:

- more than 10 days complaints at the time of screening

- a known positive SARS-CoV-2 RT PCR in the past 4 weeks

- invalid RT-PCR test results due to contamination

Next to this, the data of the following HCWs were also excluded because of known interference with the algorithm [14]:

- a deep anemia at the time of presentation to the ED (Hb < 5.5 mmol/L)

- extreme laboratory values (>10* standard deviation (SD)) in one or more of the Colab-values)

Using a standard protocol, paired naso-/oropharyngeal swabs from HCWs were collected using sterile flocked E-swabs and placed both in one sterile tube containing viral transport medium. Next to this, blood was collected in heparin- and EDTA-anticoagulated blood containers. The samples were transported to the central laboratory and immediately prepared for analysis.

### Laboratory measurements

For clinical chemistry and hemocytometric analyses, heparin- and EDTA-anticoagulated venous blood samples respectively were collected. All analyses were performed at presentation. Clinical chemistry parameters (c-reactive protein (CRP), albumin, total bilirubin, alkaline phosphatase (AP), γ-glutamyl transferase (GGT) and lactate dehydrogenase (LDH)) were obtained on routine chemistry analysers from Roche (Cobas; Roche Diagnostics, Basel, Switzerland). The hemocytometric parameters (leukocytes, erythrocytes, eosinophilic and basophilic granulocytes) were derived from a complete blood count (CBC) measured on a XN-1000 (Sysmex, Kobe, Japan).

The nasopharyngeal and oropharyngeal swab samples were obtained for SARS-CoV-2 detection using multiplex Real-Time Polymerase Chain Reaction (RT-PCR) using QIAsymphony DSP Virus/ Pathogen Mini Detection Kit (Qiagen Inc., https://www.qiagen.com). Both

the binary outcome of the RT-PCR (positive or negative), as well as the cycling time-value (Ct-value) in case of a positive RT_PCR were registered.

In case of a negative RT-PCR result in a HCW with persistent high suspicion for COVID-19 (e.g. suggestive symptoms without apparent alternative cause) the RT-PCR test could be repeated after 48 hr of the initial RT-PCR. The exclusion criterium of no more than 10 days complaints at the time of screening then still applies.

## CoLab-score calculation

The Colab-score is described in detail elsewhere [14]. In short, it is calculated by plugging the ten obtained laboratory measurands, next to the age of the HCW, into a formula: (– 6.885 + [erythrocytes ($*10^{12}$/L)] × 0.9379 –[leukocytes ($*10^9$/L)] × 0.1298 –[eosinophils ($*10^9$/L)] × 6.834 –[basophils ($*10^9$/L)] × 47.7 –log10([bilirubin (μmol/L)]) × 1.142 + log10([LDH (U/L)]) × 5.369 –log10([AP (U/L)]) × 3.114 + log10([γGT (U/L)]) × 0.3605 –[albumin (g/L)] × 0.1156 + [CRP (mg/L)] × 0.02560 + [age (years, 2 decimals)] × 0.002275). This results in a numeric value, called the CoLab-linear predictor. This linear predictor can be converted to a score using the cut-offs described in the original publication [14].

## Statistical analysis

Since the CoLab-score was developed to screen patients presenting at the emergency department (ED) for a possible COVID-19 infection, rather than exclude a SARS-CoV-2 infection in HCWs, the suitability for screening HCWs was investigated in this study. First, the discriminative ability of the CoLab-linear predictor was assessed by calculating the area under the ROC curve (AUC). Secondly, model calibration was visually assessed with a calibration plot where the CoLab-linear predictor was converted to the predicted probability (through the inverse logit function) and the proportion of observed outcomes was plotted versus expected probabilities [15]. A logistic regression model was fitted to the CoLab predicted probabilities to assess model calibration in terms of intercept and slope [15]. This was done by plotting the proportion of observed COVID-19 positives versus expexted probabilities. Ideally, observed proportions are equal to expected proportions, and this ideal-calibration line is shown as a straight line through the origin with a slope of 1. The logistic calibration line will be a logistic regression fit of the predicted probabilities. Using the intercept and/or slope from the logistic regression model, recalibrated probabilities were obtained and also plotted in a calibration plot. Thirdly, a cut-off for the CoLab-linear predictor was calculated to safely rule-out a COVID-19 infection in HCWs with an estimated 95% sensitivity. This was done by fitting a Gaussian to the distribution of the CoLab-linear predictor for all HCWs tested positive for SARS-CoV-2. The cut-off to safely rule out COVID-19 was chosen as the 5th percentile of the fitted Gaussian distribution. The number needed to screen (defined as the number of HCWs needed to RT-PCR test to find one positive) is calculated by dividing the total number of HCWs below the cut-off by the number of HCWs above the cut-off and tested RT-PCR positive. The fraction of HCWs falling below the cut-off was calculated to determine the potential reduction in RT-PCR tests. Confidence intervals for the Gaussian fit, 5th percentile and potential reduction in RT-PCR tests were obtained by bootstrapping and calculating the bias-corrected and accelerated bootstrap (BCa) confidence intervals (CIs). Finally, the relation between the RT-PCR CT-value and CoLab-linear predictor was plotted to determine if higher CT-values corresponded to lower CoLab-linear predictor values. All statistical analyses were performed in R version 4.0.5 [16], calibration plots were made using the rms-package [17], bootstrapping was done using the boot-package [18].

## Ethical considerations

The medical ethics committee of ZMC (Zuyderland METC Zuyd, registration nr. METCZ2021002) approved this study. Data were acquired after informed consent and obtained in accordance with the Declaration of Helsinki, version 2013. Participation in this study was voluntary, and each participating HCW obtained a hard copy of the 'test subject information sheet', in which the study is explained and were the participant has to give written consent. These signed consent forms were also signed by the study personel member responsible for the venipuncture. Each participant was also aware that they could opt out at any time. Because the study was restricted to HCWs, no minors (<18 years) were included.

## Results

In total, 775 health care workers (HCWs) were included in this study. Forty-nine out of the 775 HCWs were excluded (Fig 1). A total of 42 HCWs (5.8%) were SARS-CoV-2 RT-PCR positive.

Descriptive statistics for the 726 included HCWs, grouped by RT-PCR test result, are shown in Table 1. Age, erythrocytes, LD, AP and γGT do not show significant differences between RT-PCR positive and RT-PCR negative groups. All other variables included in the CoLab-score differ significantly between the 2 groups.

ROC-curve analysis of the CoLab-linear predictor is shown in Fig 2. The AUC of the CoLab-linear predictor in discriminating between RT-PCR positive and negative HCWs was 0.853 (95% CI: 0.801–0.904).

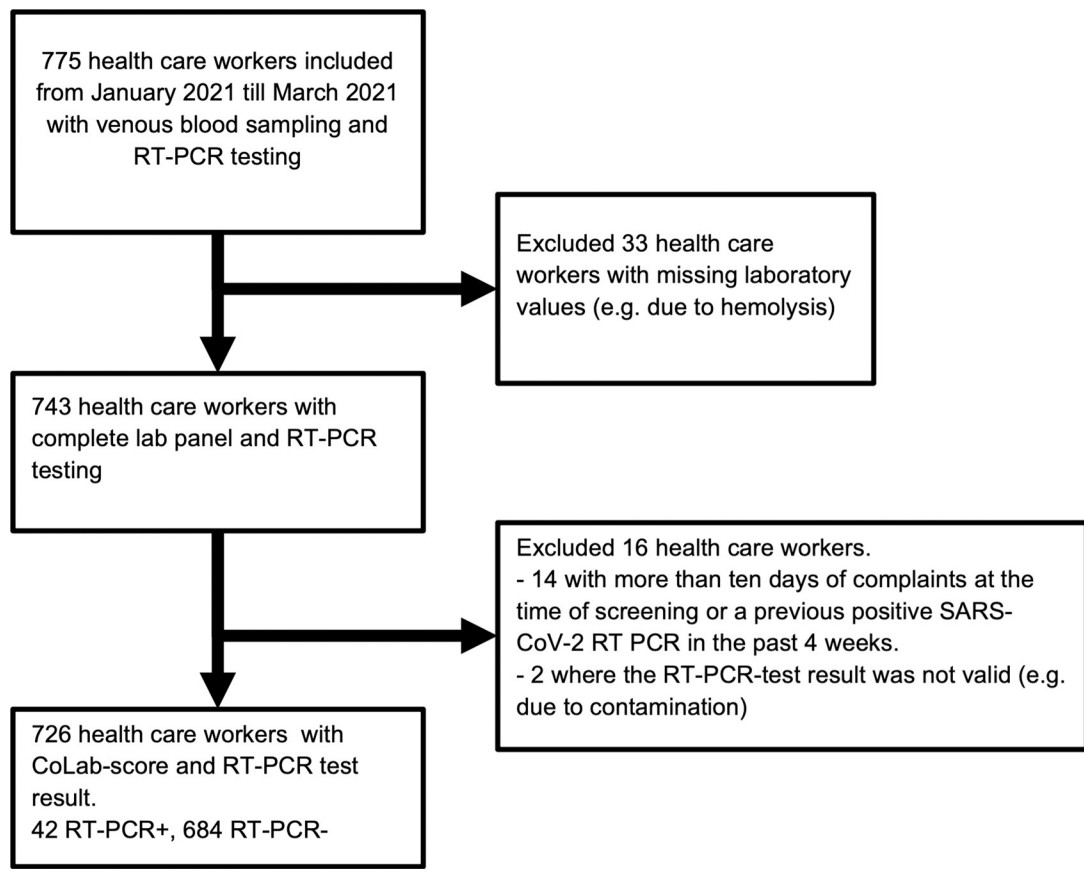

**Fig 1. Inclusion flow.**

**Table 1. Descriptive statistics.** Shown are the laboratory tests required for the CoLab-score and their mean/median results split by RT-PCR test result. For results with normal distributions, the mean value and standard deviation (in round brackets) are shown. For results that have skewed or heavy tailed distributions, the median value and the interquartile range is shown [in squared brackets]. The p-value corresponds to a t-test in cases of a normal distribution, a Man-Whitney U-test for non-normally distributed variables and a Fisher exact test for categorical variables.

| Parameter | Unit | RT-PCR negative | RT-PCR positive | p-value |
|---|---|---|---|---|
| Total number | n = | 684 | 42 | |
| Age | years | 43 (13) | 47 (12) | 0.094 |
| Gender (male) | n (%) | 105 (15.4%) | 7 (16.7%) | 0.993 |
| Erythrocytes | /pL | 4.76 (0.39) | 4.76 (0.40) | 0.897 |
| Leukocytes | /nL | 6.92 [5.68, 8.43] | 4.69 [3.90, 5.87] | <0.001 |
| Eosinophils | /nL | 0.14 [0.09, 0.21] | 0.08 [0.05, 0.12] | <0.001 |
| Basophils | /nL | 0.04 [0.03, 0.06] | 0.02 [0.01, 0.03] | <0.001 |
| Bilirubin | μmol/L | 7.0 [5.0, 9.2] | 5.0 [4.0, 6.0] | <0.001 |
| LDH | U/L | 187 (35) | 197 (42) | 0.059 |
| AP | U/L | 74.0 [61.0, 89.0] | 75.0 [64.5, 96.8] | 0.561 |
| γGT | U/L | 17.0 [13.0, 25.0] | 21.0 [14.0, 26.0] | 0.166 |
| Albumin | g/L | 45.7 (3.0) | 44.3 (2.7) | 0.005 |
| CRP | mg/L | 2 [1, 4] | 3 [1, 6] | 0.012 |

numbers between () are standard deviation; numbers between [] represents min, max.

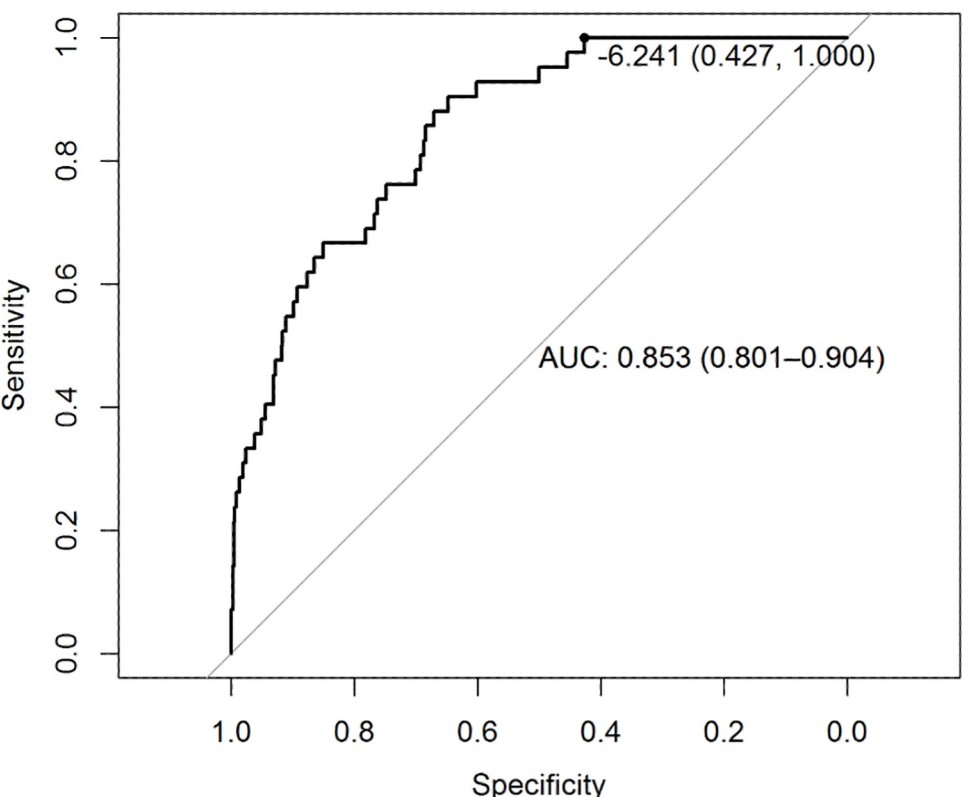

**Fig 2. ROC curve of the CoLab-linear predictor.** The area under the ROC curve is shown with the 95% DeLong confidence interval in round brackets. The displayed threshold of -6.241 corresponds to a sensitivity of 100%, i.e. no HCWs below this linear predictor were RT-PCR positive.

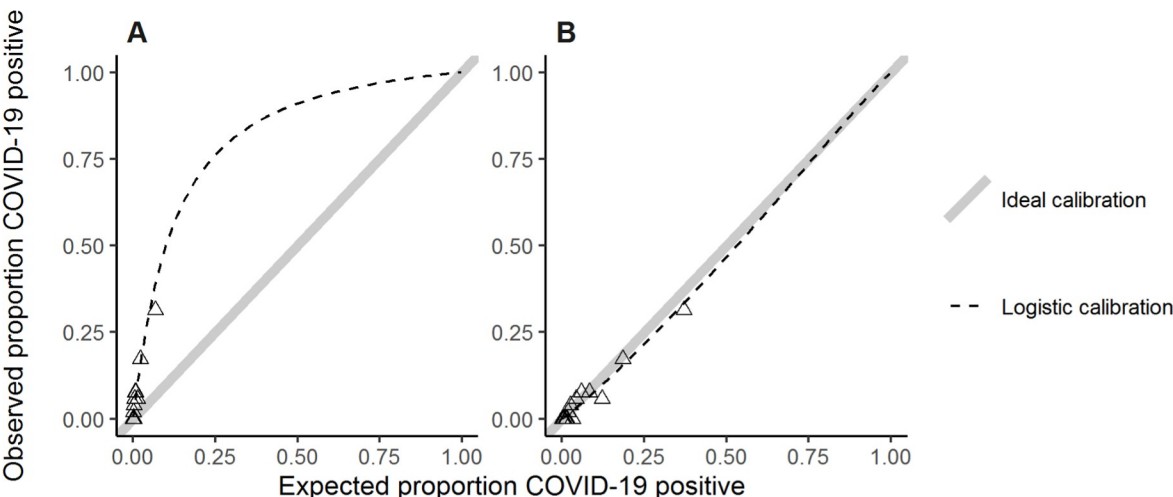

**Fig 3. Calibration plot.** A. In the calibration plot the proportion of observed COVID-19 positives versus expected proportion of positives are plotted. Observations are grouped with an average of 50 observations per group. The expected probabilities follow from applying the inverse logit function to the CoLab-linear predictor. If the observed proportion in an external dataset is lower than the expected proportion, this means risks are over-estimated, if the observed fraction is higher, risks are under-estimated. Ideally, observed proportions are equal to expected proportions, this ideal-calibration-line is shown as a straight line through the origin with a slope of 1. The logistic calibration line is a logistic regression fit of the predicted probabilities. B. Using the intercept and/or slope from the logistic regression model, recalibrated probabilities were obtained and plotted in a second calibration plot.

The calibration plot corresponding to the predicted probabilities and observed proportion of RT-PCR positives is plotted in Fig 3A. The logistic regression calibration slope is equal to 1.056 (SE: 0.1438) and the intercept 2.322 (SE: 0.6197). This implies that predicted probabilities are systematically too low but re-calibration is straightforward as there is no evidence that the slope is $\neq 1$, hence only the intercept term needs to be added to the original CoLab-linear predictor to obtain a re-calibrated linear predictor suitable for screening HCWs. The re-calibrated calibration plot is show in Fig 3B. This also illustrates that the discriminative ability of the CoLab-linear predictor is preserved but that thresholds for screening HCWs should be lower than ED patients.

To define a safe cut-off for excluding COVID-19 in HCWs, a Gaussian is fitted to the distribution of CoLab-linear predictor of HCWs that were tested RT-PCR positive (Fig 4). The Shapiro-Wilk test showed no evidence of non-normality (P-value = 0.621). The 5th percentile of the Gaussian fit of the CoLab-linear predictor is equal to -6.525 (95% CI: -7.147 to -5.999), which is recommended as the cut-off below which COVID-19 can be safely ruled-out in HCWs. Using the -6.525 cut-off, the percentage of HCWs that can be safely excluded is 34% (95% CI: 21 to 49%), with a specificity of 34%, a sensitivity of 100%, a positive predictive value of 9% and a negative predictive value of 100%. The number need to test is 12 (95% CI: 10 to 14).

In Fig 5 the relationship between the CoLab-linear predictor and the RT-PCR CT value is plotted. The fitted smooth in Fig 5 shows a rising CT value (implying a decreasing amount of template) near the lower end of the CoLab-linear predictor.

## Discussion

In this prospective study among healthcare workers (HCWs), it was shown that the model behind the CoLab-score could be used to safely exclude COVID-19 in HCWs. The original cut-off for the CoLab-linear predictor was adapted for excluding COVID-19 in HCWs. Using this adapted cut-off, a NPV of 100% was found with a specificity of 34% (95% CI: 21 to 49%). The number needed to screen by using the CoLab guided SARS-CoV-2 RT-PCR testing was

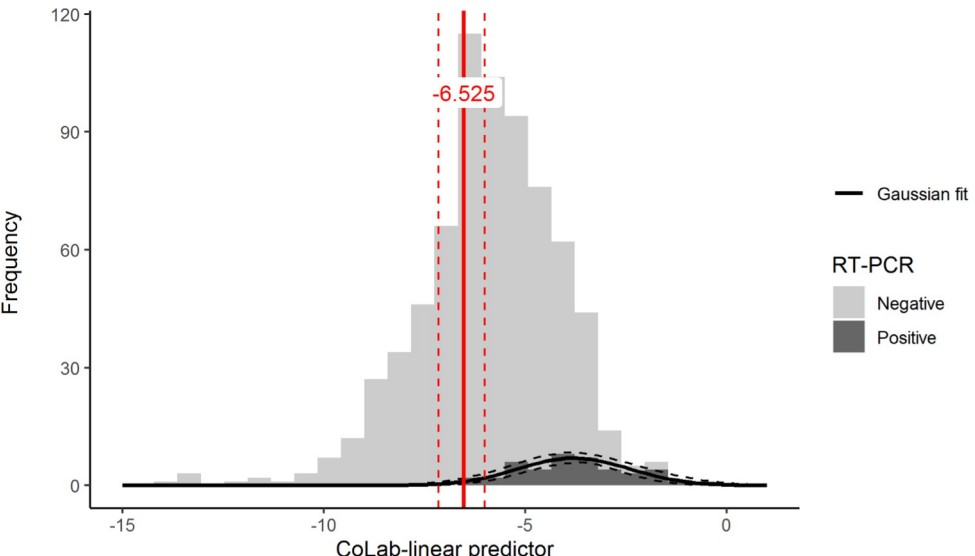

**Fig 4. Histograms and fitted Gaussian distribution of the CoLab-linear predictor split by RT-PCR result.** A normal distribution was fitted to the RT-PCR negative group (mean: -6.04, SD: 1.73), the dashed lines represent the 95% CI. The 5[th] percentile of the Gaussian distribution is shown in red and dashed lines represent the 95% CI. Linear predictor values below this 5[th] percentile are regarded as non-COVID-19.

12 (95% CI: 10 to 14). The overall prevalence of COVID-19 in this group of HCWs was 5.8%. In 33 cases (4.3%) the CoLab score could not be calculated. This was due to haemolysis of the blood sample, caused by an improper venipuncture.

RT-PCR-based methodologies are the gold standard for confirming COVID-19. There are several factors that can contribute to false-negative results, including the adequacy of the specimen collection technique, time from exposure and specimen source. Furthermore, current

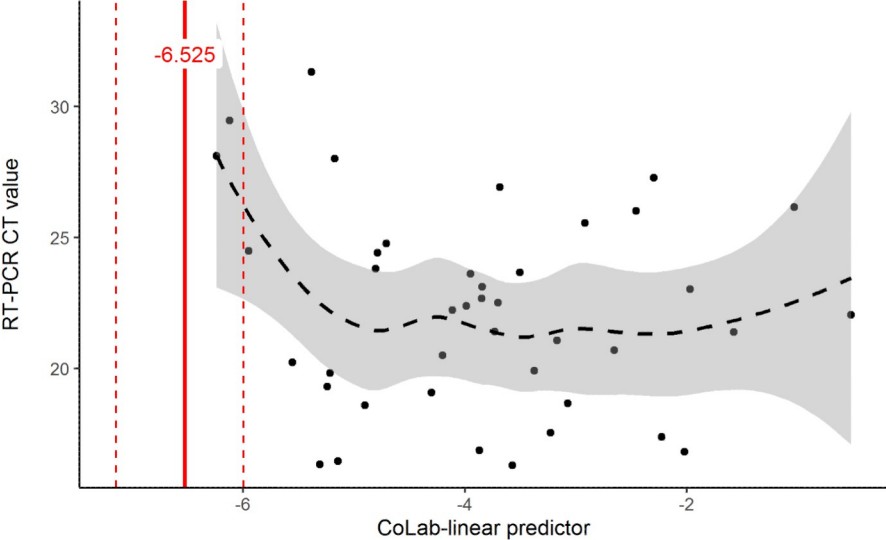

**Fig 5. CoLab-linear predictor versus RT-PCR CT value.** The CoLab-linear predictor is plotted versus the RT-PCR CT value. The red line is the CoLab-linear predictor cut-off below which HCWs are regarded as non-COVDI 19, the dashed red lines represent the 95% CI of the cut-off. The dashed line is a LOESS smooth where the 95% confidence interval is shown in gray.

and future viral changes could affect viral based diagnostics [19–21]. In addition, several studies have already shown that COVID-19 is characterized by biochemical as well as haematological changes in peripheral blood [22–25]. Next to focusing on the viral response (RT-PCR), investigating the host immune response by analysing biochemical and haematological changes in peripheral blood is an attractive alternative method [26]. As shown in the living systematic review from Wynants et al [27], a considerable number of prediction models for COVID-19 have been published until recently, and biochemical and haematological parameters are often an important part of these prediction models.

Recently the CoLab-score was developed and externally validated by Boer et al [14] to exclude COVID-19 in patients presenting at the Emergency Department, using an adaptive LASSO-regression technique [28]. This score is based on 6 biochemical and 4 haematological parameters, next to the age of the patient. It appears that the strength of the high NPV derived from this algorithm is driven by the absence of specific COVID-19 related biochemical and haematological changes in peripheral blood.

As the CoLab-score is based on a categorization of the underlying continuous linear predictor, the cut-offs define the diagnostic properties of the individual scores. Our results show that the discriminative ability of the CoLab-linear predictor is preserved when screening HCWs instead of patients presenting at the emergency department, as indicated by the AUC and calibration slope. The AUC is lower than the development group of the original CoLab study [14], but similar to the AUC reported for external datasets in the original CoLab study. Therefore, the discriminative ability seems to be preserved when classifying HCWs.

The cut-offs defined in the original CoLab publication are however not suitable for excluding COVID-19 in HCWs. This is confirmed by the calibration intercept which shows that probabilities predicted by the original CoLab-linear predictor are systematically too low for HCWs. As reliable exclusion of COVID-19 is of utmost importance in screening HCWs, a logical choice would be to select the highest cut-off with 100% sensitivity. However, one might speculate that when sampling more HCWs, the sensitivity could drop, and RT-PCR positive HCWs could occur even below this threshold. Therefore, the Gaussian distribution was fitted to the data and the 5$^{th}$ percentile chosen as safe cut-off. Doing so, the reliability increases at the expense of the number of "negative" results. We recommend that the optimal cut-off value is -6.525, where COVID-19 could be excluded in about 34% of the HCW's. Furthermore, Fig 5 suggests that potentially "missed" COVID-19 HCWs might have relative high CT values, potentially resulting in lower disease burden and contagiousness. It must be kept in mind that this study was performed in a time that the prevalence of COVID-19 was high (>10%) and no one want to classify a possible COVID-19 positive HCW as false-negative. For that reason, the optimal cut-off concerns now a very save cut-off. It can be hypothesized that when COVID-19 prevalence drops, the cut-off value can be adapted by investigating which negative predictive value can be allowed in this new setting.

It turned out that the cut-off to be used in the CoLab-algorithm in the HCW screening setting is different from that of patients presenting at the ED. This could be explained by the fact that in the screening setting the duration of the infective period is shorter, the complaints are milder and consequently the host-immune response also less pronounced [26]. In addition, in our study group there were 2 HCWs in which the SARS-CoV-2 RT-PCR was initially negative, while the CoLab-score was not negative. A week later both HCWs had a retest because of persistent COVID-19 related complaints. At that time the outcome of the linear predictor had worsened, and at that time the repeated RT-PCR test turned out to be positive. Because these 2 HCWs had more than 10 days COVID-19-related complaints, they were excluded from this study. It seems that the outcome of the CoLab-score is dynamic and follows the host immune response.

At this time, also so-called lateral flow tests (LFTs) are available which detects the presence of SARS-COV-2 antigen. They are widely adopted In ED's because of their ease of use and the rapid result (<30 minutes). However, compared to these LFTs which provide a dichotomous results, the CoLab-score provides a continuous score. Using the above-mentioned cut-off value, the CoLab-score offers a higher sensitivity and are therefore more suitable to rule-out COVID-19 than a LFT, which are only moderately sensitive [29,30].

A limitations of this study is that this study has been performed in a period where only the original SARS-CoV-2 next to only the alpha-variant of it were present. Since the CoLab-score reflects the host-response to the virus, it is expected that the accuracy of the score will not be changed by emerging SARS-CoV-2 variants. This assertion is supported by Boer et al, how found sustained diagnostic performance of the CoLab-score in periods with different dominant variants (especially Alpha- and Delta-variant) [14].

A control group of HCWs who do not have complaints or have been in close proximity of a COVID-19 patient was ideally a good control that could assist Iin the understanding for the presence of any possible that could produce false-positive results. Unfortunately, this was practically not feasible because the study was designed and performed in a period with high absenteeism of HCWs due to COVID-19. The medical board as well as the ethical committee of our hospital found it unethically to test HCWs with no complaints, especially with the knowledge that a positive SARS-CoV-2 RT-PCR test result does not always mean that the person can spread the virus actively. Because the focus of our study was directed to develop a screening method with a high negative predictive value, the missing of this control group can be remarked as a limitation of our study.

Next to this, these HCWs concerns a group of possible patients that are not registered in our hospital information system, and are treated completely anonymous for us. For that reason, we don't have information about the kind of work the HCW perfor (e.g. nurse, physician, laboratory personnel, cleaning staff, transport) and for that reason it is not possible to assess the group of HCWs who work in close proximity of a COVID-19 patient, separately from the group of HCWs who don't work in close proximity.

In conclusion, the CoLab-score is an easy and reliable algorithm that, using an adapted cut-off, can be used in screening HCWs with COVID-19 related complaints. Major advantages of this approach are that the results of the score are available within 1 hour after collecting the samples, it can be implemented in almost every hospital, even in a 24/7 setting, and the costs are minimal when compared to RT-PCR. This results in a faster return to labour process of a significant part of the COVID-19 negative HCWs (34%), next to a reduction in RT-PCR tests (reagents and labour duties) that can be saved.

## Acknowledgments

The authors would like to thank Christel Jacquot, Anke Linssen and Audrey Merry for their help with this study. Additionally, authors would like to thank the laboratory technician René Nieuwenhuysen for extracting the data out of the laboratory information systems. Finally, we appreciate the valuable assistance of the phlebotomists of the Department of Clinical Chemistry & Hematology of Zuyderland MC, Heerlen/Sittard-Geleen.

## Author Contributions

**Conceptualization:** Math P. G. Leers, Arjen-Kars Boer, Frans Stals, Henne A. Kleinveld, Dirk W. van Dam.

**Data curation:** Math P. G. Leers.

**Formal analysis:** Ruben Deneer, Arjen-Kars Boer.

**Investigation:** Math P. G. Leers, Arjen-Kars Boer, Henne A. Kleinveld, Dirk W. van Dam.

**Methodology:** Math P. G. Leers, Ruben Deneer, Arjen-Kars Boer, Henne A. Kleinveld, Dirk W. van Dam.

**Project administration:** Math P. G. Leers.

**Software:** Ruben Deneer, Arjen-Kars Boer.

**Supervision:** Math P. G. Leers.

**Validation:** Math P. G. Leers, Ruben Deneer, Arjen-Kars Boer.

**Visualization:** Ruben Deneer.

**Writing – original draft:** Math P. G. Leers, Ruben Deneer, Arjen-Kars Boer, Frans Stals, Dirk W. van Dam.

**Writing – review & editing:** Math P. G. Leers, Ruben Deneer, Guy J. M. Mostard, Remy L. M. Mostard, Arjen-Kars Boer, Volkher Scharnhorst, Frans Stals, Henne A. Kleinveld, Dirk W. van Dam.

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
