## [Decision Letter · Decision Letter 0]

22 Apr 2022

PONE-D-21-18474Use of an algorithm based on routine blood laboratory tests to exclude COVID-19 in a screening-setting of healthcare workersPLOS ONE

Dear Dr. Leers,

Thank you for submitting your manuscript to PLOS ONE. After careful consideration, we feel that it has merit but does not fully meet PLOS ONE’s publication criteria as it currently stands. Therefore, we invite you to submit a revised version of the manuscript that addresses the points raised during the review process.

We look forward to receiving your revised manuscript.

Kind regards,

Siddhartha Pratim Chakrabarty

Academic Editor

PLOS ONE

Journal Requirements:

2. In your ethics statement in the manuscript Methods and the online submission form, please state whether participation in the study was voluntary, and whether participants were aware that they could opt out at any time.

Reviewers' comments:

Reviewer's Responses to Questions

**Comments to the Author**

1. Is the manuscript technically sound, and do the data support the conclusions?

Reviewer #1: Yes

Reviewer #2: Yes

2. Has the statistical analysis been performed appropriately and rigorously? 

Reviewer #1: Yes

Reviewer #2: Yes

3. Have the authors made all data underlying the findings in their manuscript fully available?

Reviewer #1: No

Reviewer #2: Yes

4. Is the manuscript presented in an intelligible fashion and written in standard English?

Reviewer #1: Yes

Reviewer #2: Yes

5. Review Comments to the Author

Reviewer #1: Thank you for the opportunity to review the manuscript. The theme is relevant, and the article is well written.

There are some comments and questions:

Abstract

#1 - The authors should indicate the study's design with a commonly used term in the title or the abstract and inform that the data were prospectively collected.

#2 - The authors should indicate the cut-off value for excluding COVID-19 with a sensitivity of 100% in the results.

#3 - The 95% confidence interval for specificity should be included.

Methods

#1 - Since the laboratory data and outcome analysis were evaluated simultaneously, should the study be characterized as a cohort?

#2 - How many healthcare workers were not included due to not having all the laboratory data needed to calculate the CoLab-score? Are they only the 33 health care workers with missing laboratory values due to hemolysis?

#3 - Invalid RT-PCR test results due to contamination should be included in the exclusion criteria.

#4 - The RT-PCR CT-value should be explained in the laboratory measurements.

#5 - Was there a protocol to repeat the RT-PCR test with an initial negative test and high suspicion for COVID-19 remains (e.g., suggestive symptoms without apparent alternative cause)?

#6 - The authors should explain how the logistic regression model was fitted in more detail.

#7 - How many professionals did the RT-PCR due to COVID-19 related complaints and to be in close proximity to a person with COVID-19?

#8 - In addition to the analysis of overall healthcare workers, it is possible to assess these groups separately since the pre-test probabilities for the two conditions are likely to be different?

Results

#1 - "The AUC is lower than the development cohort of the original CoLab study (14) but similar to the AUC reported for external cohorts in the original CoLab study. Therefore, the discriminative ability seems to be preserved when classifying HCWs" should be moved to the discussion section.

#2 - The authors should inform the 95% confidence interval of the sensibility and specificity for the CoLab’-score cut-off value and the positive and negative likelihood ratios.

Discussion

#1 - The authors stated: "Using this adapted cut-off, an NPV of 100% was found with a specificity of 34% (95% CI: 21 to 49%)". As the negative predictive value is dependent on disease prevalence, should this NPV of 100% for the CoLab-score cut-off value be maintained with the reduction of the COVID-19 prevalence?

#2 - The authors should discuss the study's limitations.

#3 - The authors should discuss the advantage of the CoLab-score compared to the SARS-CoV-2 antigen test to fastly screen health care workers with COVID-19 related complaints.

#4 - Can the accuracy of the CoLab-score be changed by emerging SARS-CoV-2 variants, such as the Omicron variant?

Data Availability:

#1 - The authors should indicate where the data underlying the findings described are fully available.

Finally, I did not thoroughly check for grammatical errors since I am not a native English user. An appropriate language reviewer should do this.

Reviewer #2: Overall the article is well written and the work is technically sound. The findings are also useful and have potential applications in the healthcare sector. There are however a couple of minor changes that could be incorporated into the article to make the work more rigorous and better understood.

1. The study is performed only on HCW who have complaints or have been in close proximity to COVID-19 patients. While this is the focus group of the study, the authors do not analyse samples or colab scores of HCWs who do not have complaints or have been in close proximity. This would've been a good control group and would've assisted in understanding the presence of any possible noise that could produce false positives or eliminate the possibility of the same. Anyway, given that there is no data available for the same, an elaborate heuristic justification of the reason for exclusion in the main text would be useful and make the study more credible.

2. The description of figure 3 could be made better with a better description of figure 3A and 3B independently in the legends to the figures.

Once these minor changes are incorporated, the manuscript will make for a good publication.

6. PLOS authors have the option to publish the peer review history of their article (what does this mean?). If published, this will include your full peer review and any attached files.

Reviewer #1: No

Reviewer #2: No

---

## [Author Response · Author response to Decision Letter 0]

16 May 2022

Rebuttal letter PONE-D-21-18474:

Use of an algorithm based on routine blood laboratory tests to exclude COVID-19 in a screening-setting of healthcare workers

We have downloaded both templates and changes the style of the manuscript so that it meets now the PLOS ONE’s style requirements.

2. In your ethics statement in the manuscript Methods and the online submission form, please state whether participation in the study was voluntary, and whether participants were aware that they could opt out at any time.

We have added this to the Methods section, so that it is now clear that participation is fully voluntary and also that they have in advance knowledge to refuse participation to this study (rule nr 261-266).

We have completed the ethics statement in the Methods and online submission information part, by adding that each participant obtained a participation information sheets, in which the study was explained and were the participant has to give written consent. These signed consent forms are also signed by one of the study personel members responsible for the blood drawn. These double-signed forms will be saved according to the Dutch research rules (sentence nr 261-266).

Due to legal restrictions on sharing the de-identified data, individual level data cannot be shared on a public repository. However, we will make these data available for individuals who want access the data for scientific and/or academic research purposes and are willing to commit to handling the data in a manner which is consistent with confidentiality requirements. The criteria for access would broadly incorporate requests from individuals with credible academic/research credentials. Requests for access to the data should be directed to the Ethics Committee of Zuyderland Medical Centre (METC@zuyderland.nl).

Regarding this question concerning the data-availability, we can clarify and guarantee that none of the (co-)authors are a member of the ethics committee of Zuyderland The ethics-committee of Zuyderland consist of the following persons:

- prof.dr. J.W. Greve (chair)

- dr. J. Kragten (vice-chair)

- mr. F. Stroom (secretary)

- mr C. Essed 

- dr. M. ten Hoopen

- dr. D. Wong

- dr. M. de Kruif

- dr. R. Moonen

- dr. C. van Deursen

. dr. L. Frenken

- dr. E. Bols

- dr. R. Verwey

- dr. E. van de Laar

- dr. A. van der Scheer

- drs. G. Thijssen

- ms. M. Dielis

- dr. B. Winkens

- dr. W. van Asten

- dr. A. van den Hout

The list of members of the ethics committee can also be found following this URL: www.zuyderland.nl/zuyderland/bestuur-en-organisatie/metc-z/

This webpage is only in Dutch available.

We hope that this satisfy your question regarding the independency of this persons for acces to the data, and that you can update the Data Availability statement using this information.

This is correct, and for that reason we have rephrased this sentence. These data are not immediately a core part of this study, but it endorses the dynamic adaptation of the CoLab-score that follows the host immune response. For that reason, we have removed the phrase “data not shown” (sentence nr 421-422).

We have reviewed and checked the reference list and updated them when necessary.

Reviewers' comments:

Reviewer's Responses to Questions

Comments to the Author

1. Is the manuscript technically sound, and do the data support the conclusions?

Reviewer #1: Yes

Reviewer #2: Yes

OK

2. Has the statistical analysis been performed appropriately and rigorously?

Reviewer #1: Yes

Reviewer #2: Yes

OK

3. Have the authors made all data underlying the findings in their manuscript fully available?

Reviewer #1: No

Reviewer #2: Yes

Due to legal restrictions on sharing the de-identified data, individual level data cannot be shared on a public repository. However, we will make these data available for individuals who want access the data for scientific and/or academic research purposes and are willing to commit to handling the data in a manner which is consistent with confidentiality requirements. The criteria for access would broadly incorporate requests from individuals with credible academic/research credentials. Requests for access to the data should be directed to the Ethics Committee of Zuyderland Medical Centre (METC@zuyderland.nl).

4. Is the manuscript presented in an intelligible fashion and written in standard English?

Reviewer #1: Yes

Reviewer #2: Yes

OK

5. Review Comments to the Author

Reviewer #1: Thank you for the opportunity to review the manuscript. The theme is relevant, and the article is well written.

There are some comments and questions:

Abstract

#1 - The authors should indicate the study's design with a commonly used term in the title or the abstract and inform that the data were prospectively collected.

We have added the term ‘prospectively collected’ to the Methods section of the Abstract (sentence nr 89).

#2 - The authors should indicate the cut-off value for excluding COVID-19 with a sensitivity of 100% in the results.

The cut-off value for excluding COVID-19 with a sensitivity of 100% concerns a value of -6.525 for the linear predictor and is added to the abstract (sentence nr 97).

#3 - The 95% confidence interval for specificity should be included.

The 95% confidence interval for this specificity concerns 21 to 49%. This is added to the abstract (sentence nr 98).

Methods

#1 - Since the laboratory data and outcome analysis were evaluated simultaneously, should the study be characterized as a cohort?

The reviewer has a good point, and for that reason we have deleted the word ‘cohort’ in the complete manuscript, and in some cases we have replaced it by the word ‘group’ or ‘ study group’.

#2 - How many healthcare workers were not included due to not having all the laboratory data needed to calculate the CoLab-score? Are they only the 33 health care workers with missing laboratory values due to hemolysis?

The panel of laboratory tests that had to be performed to calculate the CoLab-score is automated in our laboratory systems, and are thus measured always, except for the cases that the sample is rejected on beforehand due to hemolysis. The reviewer has right with the statement that only the 33 samples of HCW’s that were affected by hemolysis, are the one that leads to the group of HCWs that were not included due to have missing laboratory values.

#3 - Invalid RT-PCR test results due to contamination should be included in the exclusion criteria.

This is added to the exclusion criteria (sentence nr. 185).

#4 - The RT-PCR CT-value should be explained in the laboratory measurements.

A better explanation of which results are registered for the RT-PCR assays as well as an explanation for the abbreviation Ct-values, are added to the section ‘Laboratory measurements” (sentence nr 210-211).

#5 - Was there a protocol to repeat the RT-PCR test with an initial negative test and high suspicion for COVID-19 remains (e.g., suggestive symptoms without apparent alternative cause)?

There is indeed a protocol that in HCWs with persistent high suspicion of COVID-19 in combination with an initial negative RT-PCR test, the test can be repeated after 48 hr of the initial PCR-test. In such case, the exclusion criterium of not more than 10 days of complaints still applies. This explanation is added to the Materials & Methods section “Laboratory measurement” (sentence nr 212-215).

#6 - The authors should explain how the logistic regression model was fitted in more detail.

The fitting of the calibration line is now explained in more detail in the Materials & Methods section “Statistical analysis”. Also, the calibration is now explained in more detail (sentence 236-240).

#7 - How many professionals did the RT-PCR due to COVID-19 related complaints and to be in close proximity to a person with COVID-19?

This is a good question from the reviewer. However, this was not an aim of the study, and for that reason this is also not registered. The study was performed in our hospital, a large teaching hospital, with two locations, and more than 3,000 hospital employees. The group of hospital employees, or health care workers, consists of personnel of the cleaning staff, transport, nurses, physician assistants and medical specialists. These are all professionals that are in proximity of COVID-19 positive patients. The study was performed during a period in which office workers were asked to work at home. We have added a paragraph with study limitations and have made a remark about it (sentence nr 454-459).

#8 - In addition to the analysis of overall healthcare workers, it is possible to assess these groups separately since the pre-test probabilities for the two conditions are likely to be different?

See our remarks above by question 7. Unfortunately, we are not able to perform this analysis.

Results

#1 - "The AUC is lower than the development cohort of the original CoLab study (14) but similar to the AUC reported for external cohorts in the original CoLab study. Therefore, the discriminative ability seems to be preserved when classifying HCWs" should be moved to the discussion section.

This is correct, and for that reason we have moved it to the discussion section (in the paragraph discussing the AUC of the HCW group) (sentence nr. 390-396).

#2 - The authors should inform the 95% confidence interval of the sensibility and specificity for the CoLab’-score cut-off value and the positive and negative likelihood ratios.

We have added this information to the result section (sentence nr 337-338).

Discussion

#1 - The authors stated: "Using this adapted cut-off, an NPV of 100% was found with a specificity of 34% (95% CI: 21 to 49%)". As the negative predictive value is dependent on disease prevalence, should this NPV of 100% for the CoLab-score cut-off value be maintained with the reduction of the COVID-19 prevalence?

This is a good remark of the reviewer, and this is also something we discussed many times with the medical board of our hospital: there is no test with a NPV of 100%, and when you collect many samples, the NPV will slowly drop. But how low may the NPV drop: is 99% acceptable, or 98%... In practice, no physician wants to give an answer to this question, especially in times when COVID-19 prevalence is high. By adjusting the CoLab-score cut-off value, the NPV will change. We have added a new paragraph to the discussion section in which we describe this clinical challenge (sentence nr. 408-413).

#2 - The authors should discuss the study's limitations.

We have added a paragraph with study limitations, just before the conclusions (with strengths). (sentence nr. 435-456).

#3 - The authors should discuss the advantage of the CoLab-score compared to the SARS-CoV-2 antigen test to fastly screen health care workers with COVID-19 related complaints.

We have added a new paragraph in the Discussion where we discuss the advantage of the CoLab-score. (Sentence nr. 424-434).

#4 - Can the accuracy of the CoLab-score be changed by emerging SARS-CoV-2 variants, such as the Omicron variant?

Since the CoLab-score reflects the host-response to the virus, it is expected that the accuracy of the score will not be changed by future SARS-CoV-2 variants. This assertion is supported by findings of Boer et al how found sustained diagnostic performance of the CoLab-score in periods with different dominant variants (alpha- and delta-variant; unfortunately, still not the omicron-variant). This explanation is also added to the Discussion section (sentence nr 435-440).

Data Availability:

#1 - The authors should indicate where the data underlying the findings described are fully available.

Due to legal restrictions on sharing the de-identified data, individual level data cannot be shared on a public repository. However, we will make these data available for individuals who want access the data for scientific and/or academic research purposes and are willing to commit to handling the data in a manner which is consistent with confidentiality requirements. The criteria for access would broadly incorporate requests from individuals with credible academic/research credentials. Requests for access to the data should be directed to the Ethics Committee of Zuyderland Medical Centre (METC@zuyderland.nl).

Finally, I did not thoroughly check for grammatical errors since I am not a native English user. An appropriate language reviewer should do this.

This has been performed by an external organization specialized in editing manuscripts for this reason.

Reviewer #2: Overall the article is well written, and the work is technically sound. The findings are also useful and have potential applications in the healthcare sector. There are however a couple of minor changes that could be incorporated into the article to make the work more rigorous and better understood.

1. The study is performed only on HCW who have complaints or have been in close proximity to COVID-19 patients. While this is the focus group of the study, the authors do not analyze samples or colab scores of HCWs who do not have complaints or have been in close proximity. This would've been a good control group and would've assisted in understanding the presence of any possible noise that could produce false positives or eliminate the possibility of the same. Anyway, given that there is no data available for the same, an elaborate heuristic justification of the reason for exclusion in the main text would be useful and make the study more credible.

We totally agree with this reviewer, and indeed a control group of HCWs who do not have complaints or have been in proximity of a COVID-19 patient was ideally a good control that could assist Iin the understanding for the presence of any possible that could produce false-positive results. Unfortunately, this was practically not feasible because the study was designed and performed in a period with high absenteeism of HCWs due to COVID-19. The medical board as well as the ethical committee of our hospital found it unethically to test HCWs with no complaints, especially with the knowledge that a positive SARS-CoV-2 RT-PCR test result does not always mean that the person can spread the virus actively. Because the focus of our study was directed to develop a screening method with a high negative predictive value, the missing of this control group can be remarked as a limitation of our study. We have added this reason for exclusion as an additional paragraph to the discussion section (sentence 441-450).

2. The description of figure 3 could be made better with a better description of figure 3A and 3B independently in the legends to the figures.

The description in the legend of figure 3 is adapted and is made independently for figure 3A and 3B (sentence nr 319-329)

Once these minor changes are incorporated, the manuscript will make for a good publication.

Thank you for this remark.

6. PLOS authors have the option to publish the peer review history of their article (what does this mean?). If published, this will include your full peer review and any attached files.

If you choose “no”, your identity will remain anonymous, but your review may still be made public.

Do you want your identity to be public for this peer review? For information about this choice, including consent withdrawal, please see our Privacy Policy.

Reviewer #1: No

Reviewer #2: No

---

## [Decision Letter · Decision Letter 1]

14 Jun 2022

Use of an algorithm based on routine blood laboratory tests to exclude COVID-19 in a screening-setting of healthcare workers

PONE-D-21-18474R1

Dear Dr. Leers,

We’re pleased to inform you that your manuscript has been judged scientifically suitable for publication and will be formally accepted for publication once it meets all outstanding technical requirements.

Kind regards,

Siddhartha Pratim Chakrabarty

Academic Editor

PLOS ONE

Additional Editor Comments (optional):

Reviewers' comments:

Reviewer's Responses to Questions

**Comments to the Author**

1. If the authors have adequately addressed your comments raised in a previous round of review and you feel that this manuscript is now acceptable for publication, you may indicate that here to bypass the “Comments to the Author” section, enter your conflict of interest statement in the “Confidential to Editor” section, and submit your "Accept" recommendation.

Reviewer #1: All comments have been addressed

Reviewer #2: All comments have been addressed

2. Is the manuscript technically sound, and do the data support the conclusions?

Reviewer #1: Yes

Reviewer #2: Yes

3. Has the statistical analysis been performed appropriately and rigorously? 

Reviewer #1: Yes

Reviewer #2: Yes

4. Have the authors made all data underlying the findings in their manuscript fully available?

Reviewer #1: Yes

Reviewer #2: No

5. Is the manuscript presented in an intelligible fashion and written in standard English?

Reviewer #1: Yes

Reviewer #2: Yes

6. Review Comments to the Author

Reviewer #1: Thank you for addressing the comments. The authors have made appropriate adjustments to the original submission which have refined and strengthened it to good effect. All my comments have been answered, and I have no further recommendations.

Reviewer #2: (No Response)

7. PLOS authors have the option to publish the peer review history of their article (what does this mean?). If published, this will include your full peer review and any attached files.

Reviewer #1: **Yes: **Fabio Ferreira Amorim

Reviewer #2: No
